# A Neo-Confucian Definition of the Relationship between Individuals and Community in the Song–Ming Period (960–1644): Start with the Discovery of Multifaceted Individuals

Meihong Zhang 

School of Philosophy and Sociology, Lanzhou University, Lanzhou 730000, China; zmh@lzu.edu.cn

**Abstract:** Alasdair MacIntyre doubts that Confucianism can discuss the relationship between individuals and community because he maintains that it is impossible to discuss the topic in depth without a Western conception of individual rights. In this article, I show that Neo-Confucianism pays extensive attention to the relationship between individuals and community by working through several Chinese thinkers' theories from the 11th to the 17th centuries. Neo-Confucianism seems to be focused on the exploration of the common principles of a community, but its real intention is ensuring the fundamentality of individual selves and making up for limitations caused by an excess of individual limitations. Thus, a new relationship is formed between individuals and community; that is, all individuals are equal and the common principles of community are independent of any individual. In order to make each individual harmonize with common principles, some mainstream Neo-Confucian thinkers attached great importance to the effort (*gongfu* 工夫) of "eliminating personal desires" (*qu renyu* 去人欲) since they thought that personal desires represented a selfish appeal that contradicts common principles. Influenced by this line of thinking, Neo-Confucianism fell into the predicament where individuals were suppressed, but this shortcoming was corrected in its later stage by defending the right to satisfy individual desires for survival. This study shows that Neo-Confucian discourse has given much thought to the problem of the relationship between individuals and community.

**Keywords:** Neo-Confucianism; individual; community; self; others; principle; desire

## 1. Introduction

How does Confucianism approach the relationship between individuals and community? Many discussions on this topic have been held from the perspective of comparative philosophy in academic circles since the beginning of the new century. The book *Confucian Ethics: A Comparative Study of Self, Autonomy, and Community*, edited by Kwong-loi Shun and David B. Wong, is worthy of special mention. The book brings together discussions among scholars from English-speaking countries who specialize in comparative philosophy between China and the West. Alasdair MacIntyre's comments are attached at the end of this book. MacIntyre not only fully affirms these philosophers' research from the perspective of comparative philosophy but also wonders about the Confucian discussion about the relationship between individuals and community, on the grounds that it is impossible to discuss the topic in depth "without making any use of any Western conception of individual rights." (MacIntyre 2004, p. 211) Coincidentally, MacIntyre's concern is indirectly responded to in Henry Rosemont Jr.'s interpretation of "what it is to be a human." According to that interpretation, "for the Confucians there are only interrelated persons, no individual selves." (Rosemont 2015, p. 93).

Thus, the problem is somewhat tangled. Does Confucianism value individuals or not? If the subject matter is confined to the pre-Qin Confucian texts, it will undoubtedly be difficult to answer this question, but if we expand our scope to include Neo-Confucianism,

which began in the Song Dynasty and dominated Chinese thought for several centuries afterwards, then we might gain a full and accurate understanding of it. Therefore, this article selects several representative thinkers from the Neo-Confucian tradition spanning from the 11th to the 17th centuries, and, by sorting out their positions regarding the relationship between individual and community, it shows the deep diversity of thought on and deep concern for the relationship between individuals and community in Neo-Confucianism, thereby addressing MacIntyre's concerns.

## 2. Shao Yong: The Discovery of Individual Selves and the Overcoming of Their Limitations

The emphasis of Neo-Confucianism on the relationship between individuals and community can be traced back to Shao Yong's 邵雍 (1012–1077) discovery of equal individuals and his overcoming of limitations caused by indulging individual selves. Before that, traditional Confucianism proposed to bridge the gap between selves and others with the rule of subjective consciousness, which is summarized by Confucius as "wishing himself to be established, sees that others are established, and wishing himself to be successful, sees that other are successful."[1] (Analects of Confucius 6: 30) Although this expresses the sincere feelings of an individual, it still cannot bridge the inner tension between selves and others, because both self and other are different individuals. If one is asked to widen his own circle of care to others' self-fulfillment, one will undoubtedly fall into the dilemma of subjectivism, thereby causing the original relationship of equality between selves and others to collapse. After all, in the era when Confucianism prevails, there may still be others who care about neither self-establishment nor self-success. What reason, then, can be available for Confucianism to change the consistent style of that group?

In order to explain the equal relationship among different individuals, Shao Yong creatively proposed the concept of "observing things" (*guanwu* 觀物). ([Shao 2010](), p. 49) In "observing things," Shao Yong first points out that equality is essential for each individual by investigating the most general feature of individuals as "things," their actual being. As for the actual beings of "things," Shao Yong believes that according to common sense, they can be divided into living beings and non-living beings. Although the actual being of each "thing" is complex and diverse, each specific being is a part of the collection of all things. Accordingly, as the most intelligent being among all "things," humans are also a part of the "many" beings constituting all "things." With the establishment of the equal relationship between humans and "things," Shao Yong also established the equality of self and other in the same way. In Shao Yong's words: "Each self is a somebody else when taking others as reference, and each other also has a self when taking himself as reference, on the ground that both self and other belong to the being of things."[2] ([Shao 2010](), p. 49) Therefore, as different individualized beings, humans are born equal, and this equality is determined by the reality of their being as "things."

Although humans are equal due to the reality of their being as "things," this does not mean they are isolated from each other in daily life. On the contrary, they always live in many given groups at once. Shao Yong believes that these groups are either consortiums of division of work made up of "scholars, peasants, craftsmen and businessmen" (*shi nong gong shang* 士農工商) or consortiums of political classes including "emperors, kings, monarchs and counts" (*huang wang di bo* 皇王帝伯). ([Shao 2010](), p. 333) Both have a normative effect on individuals in reality that is similar to that possessed by modern communities. In terms of a causal analysis, the reason why individuals are regulated by the principles of communities is not only to ensure good cooperation between self and others but also to avoid the defects of the blind subjectivity caused by indulging individual selves. Shao Yong himself is blunt about the limitations of individual selves: "Indulging individual selves makes personal feelings spread unchecked, which leads to the blindness of minds, and in turn leads to confusion about the distinction between right and wrong."[3] ([Shao 2010](), p. 152) In this sense, it is necessary for all individual selves to protect their equality and independence, as well as to avoid defects of the blinded subjectivity caused by individualism. Only in this

way can an orderly relation of cooperation be established between self and others. This also means that the tension between the self as an independent individual and the common principles of the community is also ruled out. Fundamentally speaking, common principles are not a superfluous existence but indispensable to a community as the ground rules or standards that require either the self to behave well to others or others to behave well to the self. While regulating the behaviors of all individuals in the community, these principles also guarantee their equal rights.

The appeal of the philosophy of "observing things" is not only that it reveals the equality and independence of individual selves, but it also supplies a series of constructive plans for an orderly cooperation between self and other. In Shao Yong's illustration, these plans are not theoretical assumptions but instead have long been implemented by the sages of the past. How are sages able to do that? Shao Yong thinks that it is only "by using the matrix" of "observing things on the basis of things" (*yi wu guan wu* 以物觀物). (Katz 2013, p. 155) Under the condition of "observing things on the basis of things," sages obviously go beyond the habitual matrix of "observing things from the perspective of the observer self" (*yi wo guan wu* 以我觀物) (Shao 2010, p. 49) that is common to ordinary individuals. Hence, sages can fully understand the being of others as equal and independent individuals, and then, on the premise of equality and independence, unify others and selves. In this sense, what sages think about and are concerned with can cover the thoughts and concerns of all individuals. In Shao Yong's words, it is only sages that can "use the eyes of all individuals as their own eyes," "use the ears of all individuals as their own ears," "use the mouths of all individuals as their own mouths," and "use the minds of all individuals as their own minds."[4] (ibid). Thanks to this breadth of vision, the sages' pursuits undoubtedly represent the common pursuits of all individuals, and their considerations for self and others correspondingly become a common principle of the community.

For Shao Yong, the belief that only sages can use the matrix of "observing things on the basis of things" is supported by the reality that "all things are different in size, and all individuals are different in virtue or ignorance."[5] (Shao 2010, p. 48) In that sense, the difference in individual ability has an ontological basis, and no one can deny it at any time. Although sages are much higher than ordinary individuals in ability, it does not mean that there exists an unequal relationship between them. On the contrary, Shao Yong insists sages have exactly equal status with others because, in the philosophy of "observing things," "individuals are a part of the collection of all things, and sages are a part of the collection of all individuals."[6] (Shao 2010, p. 7) In other words, whether sages or other individuals, as a part of the collection of things, they are exactly equal to each other. Even if sages can establish common principles for all individuals by virtue of their omnipotent ability, when faced with these principles, they must adhere to them unconditionally just like other individuals, and never place themselves above common principles.

### 3. The Cheng Brothers: The Independence of Common Principles from All Individuals

With Shao Yong's establishment of common principles, what Neo-Confucianism needed to do next was to reveal how they play a normative role in the daily lives of all individuals. In fact, not long before the rise of Neo-Confucianism, when Buddhism and Daoism criticized Confucian ethical principles as being centered on ruler/minister and father/son relations, this problem had already become a topic of hot debate. For example, the viewpoint of "going beyond Confucian ethical dogmas and conforming to nature"[7] (Ji 2014, p. 402) in Wei-Jin Neo-Daoism and the idea of "Buddhists disrespecting the king"[8] (Shi 1992, p. 220) in the Buddhist tradition posed a serious challenge to the universality of Confucian ethical principles. To some extent, elucidations on the universality of common principles in Neo-Confucianism were a unified response to the criticisms against Confucian ethical principles by Buddhism and Daoism. Among them, the "theory of heavenly principles" (*tianli lun* 天理論) which was put forward by the Cheng brothers, Cheng Hao 程顥 (1032–1085) and Cheng Yi 程頤 (1033–1107), shows very profound insight.

It should be pointed out that in explaining the normative role of common principles in individual daily life, the "theory of heavenly principle" does not offer a perspective from social ethics but instead an ontological one: "All things under heaven can be seen in the light of principle. As long as there is a thing, there must be a law, and everything has its principle."[9] (Cheng and Cheng 2004, p. 193) This is Cheng Yi's explanation for the combination of things and principle. In his view, principle is "ontologically prior to things," and "it explains not only how a thing exists but also why a thing is such a particular thing instead of something else." (Huang 2014, p. 201) Based on the above elaboration, Cheng Yi clarifies the necessity of principle as a natural law and establishes the inevitability of principle as a moral law as well. However, how can Cheng Yi do so? It is because, for him, the statement that "all things under heaven can be seen in the light of principle" already contains concern about human beings and, compared with things, human beings have a duality; that is, human beings are not just a thing that accords with natural principles but they are also beings that accord with moral principles, too. Thus, an inevitable unity is formed between human beings and common principles that is rooted in moral law, and as long as there are human beings, there must be common principles, too. In the Cheng brothers' understanding, this kind of unity is unconditional, and "Even in times of difficulty and restlessness, it be this way."[10] (Cheng and Cheng 2004, p. 38)[11]. From what the Cheng brothers say, it can also be concluded that common principles, including Confucian ethical principles, are inevitable for any individual, and no one can escape the restriction of these principles. What is more, this is presented in terms of the unity of individuals and Confucian ethical principles, not as an assumption but as something founded on an ontological fact; that is, on the fact that "there is not a single one of the ten thousand things and the many affairs that does not each have its own proper place."[12] (Cheng and Cheng 2004, p. 968).

To some extent, the unity of individuals and Confucian ethical principles is different from that of things and natural principles. The former involves an "ought to be" while the latter is keen on questioning a "to be." Therefore, to say that the relation between individuals and Confucian ethical principles is inevitable is just a necessity within the pursuit of a universal moral ideal, but the relationship between things and principle provides a certain ontological guarantee. The Cheng brothers were aware of this difference and, when further discriminating the relationship between the two kinds of principles, Cheng Yi specifically mentioned the method of "analogy." (*leitui* 類推) (Cheng and Cheng 2004, p. 157) In such an "analogy," the clarification of the ontological relationship of things and natural principles is the premise and, through this, the unity of individuals and Confucian ethical principles is revealed to also have similar features. In short, this method for discovering meaning can be summarized in two points: first, in the ontological dimension, it is an absolute truth that there is no thing under heaven that is not unified with natural principles; second, in the ethical dimension, Confucian ethical principles should be a necessity for all individuals and constitute the definition of what human beings are. Relying on the latter dimension, the Cheng brothers strongly refuted the criticisms of Confucian ethical principles by Buddhism and Daoism, which argued that Confucian ethical principles are superfluous. On that basis, the Cheng brothers asserted that any individual, whether they be Daoist or a Buddhist, must accord with this ethical principle, and this is also a truth of daily life.

In addition to providing theoretical support for explaining the inevitability of Confucian ethical principles through the method of "analogy," the Cheng brothers also discuss the objectivity and completeness of Confucian ethical principles. This so-called objectivity is first shown in terms of natural principles, which are not changed by individual will or desire. The Cheng brothers' borrow from Xunzi 荀子 (c. 313–c. 238 BCE) to say that this principle does "not appear due to Yao, nor does it disappear due to Jie."[13] (Cheng and Cheng 2004, p. 31) Compared with objectivity, completeness emphasizes that natural principles have definite connotations that do not change with the development of concrete beings. In the Cheng brothers' words, "how can we say that it appears or disappears, that

it increases or decreases? It is not originally incomplete."[14] (ibid) It can be said that it is precisely because of their objectivity and completeness that natural principles are called "heavenly principles" by the Cheng brothers. Corresponding to the establishment of the objectivity and completeness of natural principles, Confucian ethical principles centered on "benevolence" (*ren* 仁) and "righteousness" (*yi* 義) are also interpreted to have similar characteristics. This means that the Confucian ethical principles are both objective and complete. "When does it say that Yao increases the ruler's principle because of his own acting on principle of the ruler and that Shun increases son's principle because of his own acting on the principle of filiality? These principles remain as they ever were."[15] (Cheng and Cheng 2004, p. 34) It can be seen that, in the face of objective and complete Confucian ethical principles, no matter how well sages, including Yao and Shun, perform ethically,[16] they can at most only add to the number of good moral examples rather than adding a new element to the definite connotations of ethical principles. In this way, the Cheng brothers emphasized the independence of Confucian ethical principles from all individuals.

Of course, ethical principles are different from moral laws because the former are always associated with particular historical situations. To such an extent, it seems somewhat unintelligible to assert the objectivity and completeness of any ethical principle. However, in terms of responding to the criticisms of Buddhism and Daoism, and further highlighting the universal features of Confucian ethical principles, the above assertion can provide related support for the Cheng brothers in argument. It is out of these considerations that Confucian ethical principles have also been regarded as another form of "heavenly principle" by the Cheng brothers. There is no doubt that the Cheng brothers highlight the independence of Confucian ethical principles, which not only increases the common features of Confucian philosophy but also warns against the emergence of an authoritarian personality represented by sages. This idea of stressing the independence of ethical principles provides a theoretical reference for Neo-Confucianism to deal with the relationship between individuals and community.

### 4. Zhu Xi and Wang Yang Ming: Eliminating Desires to Preserve Harmony between Individuals and Community

The establishment of common principles, including Confucian ethical principles, is only a theoretical consideration that does not ensure an adequate normative effect on individuals in daily life. Neo-Confucianism also acknowledges the tension between theory and practice. In the traditional understanding of Neo-Confucianism, this tension is not caused by the flaws of ethical principles but is mainly caused by unchecked individual limitations. The Cheng brothers pay special attention to the emergence of the tension between individuals and community from the perspective of individual limitations. However, in their eyes, these limitations are obviously not those subjective ones mentioned by Shao Yong but instead are "desires" (*yu* 欲). Precisely speaking, they are "personal desires" (*renyu* 人欲) which are regarded as self-centered demands. The Cheng brothers believe that an excess of personal desires is incompatible with the common principles that are used to keep communities healthy on the grounds that "pursuing personal desires leads to little dedication, and the orderly operation of common principles is built on individual dedication. Treating others with common principles is the means by which one is morally commiserate."[17] (Cheng and Cheng 2004, p. 372) As a result, an alternative relationship between common principles and personal desires is formed. Cheng Yi succinctly summed up this relationship as "either heavenly principles or personal desires . . . once there are no personal desires there will be only heavenly principles."[18] (Cheng and Cheng 2004, p. 144)[19].

Zhu Xi 朱熹 (1130–1200) inherited the Cheng brothers' definition of the relationship between individuals and community in terms of the relationship between common principles and personal desires. Like the Cheng brothers, Zhu Xi also notices that there is an alternating relationship between common principles and personal desires, and what is more, he points out that the proliferation of personal desires lessens the harmony between

individuals and community. In his view, between individuals and community, "how can a good order and harmonious relation be possible if individuals act only out of their desires?"[20] (Li 1986, p. 606) Zhu Xi is very vigilant about the harm of excessive individual desires and even points out that "its destructive effects are strong enough to lead to a fall of the whole state."[21] (Li 1986, p. 2395) Zhu Xi stresses that, for each individual living in the community, eliminating personal desires should become part of their daily business; especially, "at the moment when heavenly principles and personal desires are in conflict, one must overcome their selfish desires in whatever situation they counter and not leave them alone casually. For this reason, one must first of all understand what common principles he should be in accordance with, and then resolutely implement those principles to eliminate personal desires."[22] (Li 1986, p. 2800).

How were principles to be employed in the elimination of personal desires? Zhu Xi taught that one needs to do so through "effort" (*gongfu* 工夫): "for the effort of those engaged in learning, there are only two things to do, which is holding on to reverence and exhaustively seeking principles."[23] (Li 1986, p. 150) Here, "exhaustively seeking principles" (*qiongli* 窮理) belongs to the first stage of effort, which includes reading the classics (*dujing* 讀經) and apprehending principles in things (*gewu* 格物). By means of "exhaustively seeking principles," individuals can deeply realize that common principles are inescapable to themselves like the inevitability of heavenly principles to things. This understanding clarifies the goal of their actions. "Holding on to reverence" (*jujing* 居敬) belongs to the second stage of effort, which is dedicated to strengthening individual autonomy. By means of "holding on to reverence", individuals internalize the inevitable common principles in their hearts, and then make these principles the master of their actions. It should be added that the first and second stages of effort only indicate the logical sequence and do not tell us that one stage is more important than the other. As far as the feasibility of this method of effort is concerned, these two stages are very important and are both indispensable: "if one does not exhaustively seek principles, then no understanding why principles are inevitable will be had . . . if one does not hold on to reverence, then no collecting principles in the heart will be had."[24] (Li 1986, p. 151) It can be seen from the above that "holding on to reverence" and "exhaustively seeking principles" cannot be separated because "exhaustively seeking principles" is not just to understand the knowledge of principles but to internalize them in the heart, which requires the intervention of reverence. In addition, "holding on to reverence" is not a pure action with no concerns; instead, it nurtures principles. On the premise that "holding on to reverence" and "exhaustively seeking principles" cannot be separated, Zhu Xi points out that "if acting with reverence, heavenly principles will prevail; otherwise, if acting without reverence, personal desires will flood forth."[25] (Li 1986, p. 287) From the above, we see that Zhu Xi has redefined effort in terms of the relationship between common principles and personal desires.

Wang Yangming 王陽明 (1472–1529) also had a profound influence on the theory of the effort of eliminating desires. Compared with Zhu Xi, Wang Yangming seems to have "little patience with lengthy, systematic approaches" such as reading the classics and apprehending principles in things, but focuses on practicing the theory of extending conscientious knowing (*zhi liangzhi shuo* 致良知説). (Angle 2009, p. 150) "On the whole, learning and effort is about paying attention to the intentions in one's head, if one focuses one's intention on extending moral learning then in whatever one hears and sees there will be nothing not the effort of extending conscientious knowing."[26] (Wang 2015, p. 88) According to the theory of extending conscientious knowing, it is the "moral knowing" of the mind-heart that determines what individuals think and do in their daily lives, including the practice of common principles and overcoming of selfish desires. In this sense, the exertion of effort in extending conscientious knowing, which is centered on expanding the dominating function of the mind-heart, is not to let individuals return to the state of no demands and ignorance but to ensure that the mind-heart can absolutely dominate over action by eliminating personal desires.

All are only from the mind-heart, and the mind-heart decides whether principles are possible. If the mind-heart is not covered by selfish desires, it will be filled with heavenly principles, and there is no need to add more from the outside. When all actions are from the mind-heart filled with heavenly principles, serving fathers must be in accordance with filial piety, and serving kings must be in accordance with loyalty, and making friends and governing people must be in accordance with faith and benevolence. It is enough for the mind-heart to put forth efforts on eliminating personal desires and keeping heavenly principles.[27] (Wang 2015, p. 3)

Obviously, Wang Yangming does not treasure the knowledgeable understanding of common principles as much as other Neo-Confucians. He instead pays more attention to doing or not doing in terms of individual actions, which are conditioned by the degree of eliminating personal desires. As long as personal desires are completely eliminated, then the individual mind-heart will naturally point to the common principles and the tension between individuals and others will thus be resolved. With the occurrence of this, there achieves a full integration between individuals and the community, in which instance, each individual appears very harmonious and calm in carrying out his daily business that he should conduct in the community. In Wang Yangming's words: "Even if one has been engaged in tedious business for the community all his life, he still does not feel tired at all, and for this reason, despite at the bottom of the community, he feels quite calm and does not think it is humble."[28] (Wang 2015, p. 67).

From the discussion of Zhu Xi and Wang Yangming on the effort of eliminating desires, it can be seen that the leading members of Neo-Confucianism attached great importance to the harmony between individuals and community. However, because they focus too much on the overcoming of individual limitations in the promotion of harmony, there is an imbalance in their dealing with the relationship between common principles and individual needs that is embodied in the emphasis on common principles and the neglect of individual needs. This suppression of personal desires obviously deviates from Shao Yong's original intention of covering the thoughts and concerns of all individuals and is not very close to the reality of daily life.

Finally, there is another detail that needs to be revealed; that is, although the leading members of Neo-Confucianism advocate the suppression of individual personal desires, they do not deny the importance of conscientious knowing in the individual mind-heart. Rather, some of them particularly emphasize the significance of maintaining the independence of the individual mind-heart in judging what is right or wrong. Among them, Wang Yangming is the most conspicuous in this regard: "Whether the teachings are important or not depends on the examination of the individual mind-heart. Even if it is a quote from Confucius, we still dare not believe it to be right when it cannot stand the examination of our mind-heart."[29] (Wang 2015, pp. 93–94) These are very unique expressions. From the perspective of the history of Confucianism, such expressions are of very considerable significance, showing the open minds of mainstream Neo-Confucianists, centered on Wang Yangming, and their beliefs against authority worship and encouraging respect for individual autonomy.

## 5. Li Zhi and Wang Fuzhi: Defending the Right to Satisfy Individual Desires

The excessive suppression of individual desires by the leading members of Neo-Confucianism led to a reactionary movement to redefine the relationship between individuals and community, within which Li Zhi 李贄 (1527–1602) seems to be the most radical. As a disciple of the Taizhou School 泰州學派 that followed the teachings of Wang Yangming, Li Zhi realized the distortion and falseness of the Neo-Confucian idea of eliminating personal desires in the relationship between individuals and others. This idea leads to there being a large number of individuals who "only know to cater to others but not to cherish themselves, and who only pursue a good reputation but not actual effects. If one entreats someone who is already like these then both will go hand in hand into a trap."[30] (Li 2009, p. 6) Besides this, Li Zhi also believes that this action that completely ignores individuals is



evidently contrary to the actual situation of human beings on the grounds that, in daily life, "seeking advantages and avoiding disadvantages is the common aspiration of all individuals."[31] (Li 2009, p. 41) It is the instinct of "seeking advantages and avoiding disadvantages" that makes individuals act toward achieving definite goals.

In view of the fact that "seeking advantages and avoiding disadvantages" is a common characteristic of human beings, Li Zhi emphasizes the positive drive of selfish desires in the pursuit of goodness in the individual mind-heart. Taking the daily life of individuals as an example, Li Zhi points out that "The existence of mind-heart must be conditioned by the existence of individual selves. If there are no individual selves, there will be no mind-heart. For example, those who are engaged in farming often think that they as individual selves will have the harvest in autumn so they always work very hard in plowing the fields."[32] (Li 1959, p. 544) It can be seen that Li Zhi especially highlights the decisive role of individual desires in understanding how the dominant function of the mind-heart is possible, and he even regards the satisfaction of desires as a common rule of individuals. Starting from this understanding, Li Zhi makes abstract ethical principles utilitarian and insists that "the issue of dressing and eating is the only concern of ethical principles; apart from dressing and eating, there are no other ethical concerns."[33] (Li 2009, p. 4) Due to the excessive emphasis on the satisfaction of individual desires, the independence of common principles is weakened, and this inevitably pushes Li Zhi's definition of the relationship between individuals and community in the opposite direction to mainstream Neo-Confucian thought.

In terms of criticism of the problems of mainstream Neo-Confucianism, Wang Fuzhi 王夫之 (1619–1692) is not as extreme as Li Zhi. He instead takes a positive approach to remedying the ills of Neo-Confucianism. Focusing on explaining the line from the *Book of Songs* (*shijing* 詩經) that says "only a full seed has vitality" (*shihan si huo* 實函斯活) (Zhou 2010, p. 486), Wang Fuzhi points out that the truth of this line can be used to clarify how individuals adhere to the common principles of the community. That is to say, between individuals and common principles, it is only when individuals have full vitality in the body that it can be possible for them to adhere to common principles. Otherwise, for any individual who lacks vitality, adhering to common principles is undoubtedly just an empty phrase. In this sense, Wang Fuzhi puts the satisfaction of individual desires for survival in a fundamental position when discussing the relationship between individuals and common principles because this is the only way to ensure that individuals are full of vitality. "Individual selves with destitute bodies are just like blighted grains with empty cores, and that makes their vitality very weak. The lack of vitality also leaves individual selves accomplishing nothing in the practice of benevolence."[34] (Wang 2011, vol. 3, p. 501).

As can be seen from the above, "Wang Fuzhi has a fundamental confirmation of what is essential to human survival" that mainly includes "physical needs and material desires." (Liu 2018, p. 270) In this regard, Wang Fuzhi not only states that all individuals cannot be without desires, but also insists that "The desires that are common (*gong* 公) to all under heaven is principle; and when each individual appropriately satisfies their own desires there is fairness (*gong* 公)."[35] (Wang 2011, vol. 12, p. 191) This does not mean that all individual desires should be averaged or unified in terms of satisfaction. Based on such an understanding of the relationship between principles and desires, Wang Fuzhi points out that no matter how cumbersome and complex principles are, they should not conflict with individual desires for survival. In other words, satisfying desires for survival is the most essential need of all individuals, and this is directly related to their existence as living organisms. Therefore, "do not establish a principle that is ultimately separated from individual desires."[36] (Wang 2011, vol. 6, p. 913) At the same time, Wang Fuzhi does not think that it is necessary to discuss the common principles of a community for an individual with no signs of life.

For Wang Fuzhi, advocating the satisfaction of individual desires for survival is not due to a need to pursue creature comforts but instead is to emphasize that we should not "treat personal desires as dangerous things such as snakes and scorpions,"[37] (Wang

2011, vol. 6, p. 675) especially when it comes to the definition of the relationship between individuals and the common principles of their community. Otherwise, if individual desires for survival are regarded as a negative thing and "completely cut off relations with others",[38] (ibid) then it will undoubtedly force individuals into a desperate situation for survival. To such an extent, the satisfaction of individual desires for survival has become a basic right, which seems both essential and reasonable. As a basic right, it means that all individuals are neither blind nor unrestrained in the satisfaction of their desires for survival but only obtain their due shares according to corresponding principles. Wang Fuzhi believes that the self-discipline shown by individuals in the pursuit of the satisfaction of their desires reflects their uniqueness as human beings: "The difference between human beings and animals is that daily physical needs of the former are always orderly met in accordance with clear principles."[39] (Wang 2011, vol. 3, p. 492) That is to say, as flesh-and-blood beings, all individuals have physical desires by birth, and as human beings, they are fully capable of controlling the degree to which they satisfy their desires to an appropriate level, thereby allowing individuals to defend their rights without hindering their adherence to the common principles of their community. Obviously, Wang Fuzhi bravely stands at the forefront of his era when it comes to discussing how a harmonious relationship between individuals and community can be established.

## 6. Conclusions

Through the analysis of the philosophy of the above seven Neo-Confucians, we have seen that there existed multidimensional discussions on the relationship between individuals and community in the Neo-Confucian tradition. These include the dimension of principles and desires, the dimension of selves and others, the dimension of the one and the many, the dimension of the common and the personal, etc. Regardless, these discussions point to the maintenance of a harmonious relationship between individuals and community. In this Neo-Confucian discourse, the reality of individuals and the independence of the common principles of community were highlighted more than ever before, and that is why Neo-Confucianism deserves to be affirmed and taken seriously regarding the topic of individual rights and the community. Of course, it is undeniable that, in promoting the harmony between individuals and community, Neo-Confucianism once made the mistake of suppressing individual needs. This imbalance of emphasizing common principles and despising individual needs led to certain criticisms by future generations of thinkers. However, it is also necessary to see that it was the deformed definition of the relationship between individuals and community by mainstream Neo-Confucian thinkers that prompted the reactionary movement of actively defending the right to satisfy individual desires for survival in the later stage of Neo-Confucianism. This reflected the fact that Neo-Confucianism had a sufficiently strong power of self-correction and, what is more, it reveals that Neo-Confucianism took seriously the human right for survival. In view of the above, we can conclude that MacIntyre's claim that Confucianism does not have a conception of individual rights is not strong enough. Especially in the work of some Neo-Confucianists, such as Shao Yong and Wang Fuzhi, plenty of counterexamples can be found it to demolish it.

**Funding:** This research received no external funding.

**Data Availability Statement:** Not applicable.

**Acknowledgments:** Grateful thanks are due to two anonymous reviewers whose comments on the original manuscript are considerable help to me.

**Conflicts of Interest:** The author declares no conflict of interest.

## Notes

1    己欲立而立人，己欲達而達人。
2    我亦人也，人亦我也，我與人皆物也。

3　任我則情，情則蔽，蔽則昏矣。

4　用天下之目為己之目;用天下之耳為己之耳;用天下之口為己之口;用天下之心為己之心。

5　物有大小，民有賢愚。

6　人亦物也，聖亦人也。

7　越名教而任自然。

8　沙門不敬王者。

9　天下物皆可以理照，有物必有則，一物須有一理。

10　顛沛造次必於是。

11　This statement is originally from the *Analects* of Confucius, and Confucius used it to reveal the relationship of exemplary persons to the principle of benevolence: "Exemplary persons will not deviate from benevolence even in the short time it takes to finish a meal, and in times of rush and restlessness must also maintain accordance with benevolence". (Analects of Confucius 4: 5).

12　萬物庶事莫不各有其所。

13　不為堯存，不為桀亡。

14　更怎生說得存亡加減？是它元無少欠。

15　幾時道堯盡君道，添得些君道多；舜盡子道，添得些孝道多？元來依舊。

16　Yao and Shun have always been regarded as ancient sages by the Confucian tradition.

17　人循私欲則不忠，公理則忠矣。以公理施於人，所以恕也。

18　不是天理，便是私欲 …… …… 無人欲即皆天理。

19　"Heavenly principles" here refer to the common principles that are independent of all individuals. In the context of Neo-Confucianism, all "heavenly principles" that appear together with personal desires refer to common/ethical principles and are always used to indicate the universality and independence of the latter.

20　只是人欲私心做得出來，安得有序，安得有和。

21　其流弊便有喪邦之理。

22　天理人欲交戰之機。須是遇事之時，便與克下，不得苟且放過。此須明理以先之，勇猛以行之。

23　學者工夫，唯在居敬、窮理二事。

24　若不窮理，又見不得道理 …… …… 不持敬，看道理便都散，不聚在這裡。

25　敬便是天理，肆便是人欲。

26　大抵學問工夫，只要主意頭腦是當，若主意頭腦專以致良知為事，則凡多聞多見，莫非致良知之功。

27　都只在此心，心即理也。此心無私欲之蔽，即是天理，不須外面添一分。以此純乎天理之心，發之事父便是孝，發之事君便是忠，發之交友治民便是信與仁。只在此心去人欲、存天理上用功便是。

28　終身處於煩劇而不以為勞，安於卑瑣而不以為賤。

29　夫學貴得之心。求之於心而非也，雖其言之出於孔子，不敢以為是也。

30　但知為人，不知為己；惟務好名，不肯務實。夫某既如此矣，又複與此人處，是相隨而入於陷穽也。

31　趨利避害，人人同心。

32　人必有私而後其心乃見。若無私則無心矣。如服田者。私有秋之獲而後治田必力。

33　穿衣吃飯，即是人倫物理；除卻穿衣吃飯，無倫物矣。

34　我體不立，則穀之仁猶空之仁，我之仁猶空之仁，蕩然不成乎我，而亦無以成乎仁矣。

35　天下之公欲，即理也；人人之獨得，即公也。

36　終不離欲而別有理也。

37　把這人欲做蛇蠍來治。

38　與他一刀兩段。

39　人之異於禽獸者，粲然有紀於形色之日生而不紊。

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
