# Peer review of "A Neo-Confucian Definition of the Relationship between Individuals and Community in the Song–Ming Period (960–1644): Start with the Discovery of Multifaceted Individuals"

_religions, doi:10.3390/rel13090789_

Round 1
Reviewer 1 Report
Summary
This article is spurred by Alasdair MacIntyre’s comment in his “Questions for Confucians: Reflections on the Essays” in Confucian Ethics: A Comparative Study of Self, Autonomy, and Community (2004). MacIntyre wonders how Confucian thinkers could engage in a deep discussion on the relationship between individuals and community, “without making use of any Western conception of individual rights.” Drawing from ideas of several neo-Confucians, the author aims to establish a distinctive Confucian discourse on this relationship. Multiple models are presented: (1) Shao Yong’s suggestion that one take others as reference in establishing the common ground; (2) the Cheng brothers’ emphasis on the collective identity of human beings and the shared heavenly principle for all people; (3) Zhu Xi and Wang Yangming’s demand of eliminating personal desires to preserve communal harmony; (4) Wang Fuzhi’s recognition of the individuals’ right to gratifying personal desires for survival and basic needs.
General Comment
This paper is well written with a clear thesis: There are ways to explore the relationships between the individual and the community without employing the Western conception of individual rights. Neo-Confucianism offers a variety of models to establish a balanced relationship between the self and the community. The author is erudite in neo-Confucianism and shows good understanding of each philosopher’s key idea relevant to the thesis. The paper is successful in presenting conceptual pluralism of the relationship between individuals and the community. The overall argument is strong, but with specific neo-Confucian philosophers’ views the defense or the textual support is not sufficient. I recommend that the author elaborate more at various places throughout the paper (see my specific comments for the author) to zoom in on the relationship between the self and others, the individual and the community. The notion of “common principle” needs to be explained from the start: what are the common principles and in what sense are they “common”? I also suggest that the author addresses MacIntyre’s statement head-on and presents an argument for why the concept of individual rights is not essential or indispensable in this socioethical context. (Or: Alternatively, the author could emphasize (as the final conclusion claims) that in Wang Fuzhi’s picture there is a hidden allusion to individuals’ rights to survival, to basic biological needs and desires, so MacIntyre was wrong in claiming that Confucianism does not have this notion.)
Specific Comments
1. Page 2, Lines 55-61: The author refers to “the rule of subjective consciousness” in traditional Confucianism as Confucius’ statement: “Let others get what you want, and let others achieve what you want.” The citation is Cheng 1990, p. 428. However, the cited source is Cheng’s commentary on the Analects in Chinese, so this English rendition must have been the author’s translation. This English rendition does not accurately express what Confucius had said. The original comment is Confucius’ definition of ‘ren’—the virtue of humaneness. A more fitting translation of that remark is: “wishing himself to be established, sees that others are established, and wishing himself to be successful, sees that other are successful.” It does not ask one to determine “what others need based on one’s own needs” (line 60); it merely asks one to widen one’s circle of care to others’ self-fulfillment. The author criticism of traditional Confucianism as “subjectivism” is unfair or needs more justification to say the least.
2. Page 3, Line 108: The author says, “In Shao Yong’s words, it is only sages that can use the eyes of all individuals as their own eyes…”. This weakens the point that the author aims to establish in this article. If only sages could accomplish this, then everyone else fails to do so. The literal translation of the original Chinese text makes the sages’ ability “godly” or omnipotent. I suggest that the author strengthens the point about Shao Yong’s recommended methodology “observing things on the basis of things” rather than the sages’ ability to “represent the common pursuits of all individuals.” For example, how does one observe things on the basis of things? What kind of mentality must one possess to do so? The author could elaborate more.
3. Page 4, Line 177: The sentence “compared with objectivity, completeness emphasizes that natural principles do not increase or decrease…” does not make sense. The meaning of ‘completeness’ is not clearly explained. On Line 172, the author says that the Cheng brothers discuss the objectivity and completeness of Confucian ethical principles. How could one ascertain that these ethical principles are either “objective” or “complete” (how could any ethical principle be complete)? We cannot simply take the Cheng brothers’ words for it. More explanation is needed for the argument.
4. Page 5, Lines 189-191: This statement is unclear: “no matter how well sages perform ethically, they can … rather than adding to the actual existence of ethical principles”. I take it that the author meant that sages do not create ethical principles since these principles are heavenly principles. I recommend rephrasing here. The claim made in this paragraph (equality for all individuals) needs more support from the text.
5. P. 6, Line 232: “in whatever situation they encounter otherwise they find themselves in excessive” is the author’s own translation of Li 1986. This sentence does not make sense.
6. P. 6, Line 243: “common principles are inevitable to themselves” sounds strange. I think the author meant “inescapable” rather than “inevitable” as these principles are preordained to us, or in us, as our intrinsic essence.
7. P. 7, Line 291-92: The author claims that “it can be seen that the leading members of Neo-Confucianism attached great importance to the harmony between individuals and community.” However, in this section the author did not explain how eliminating selfish or personal desires can enhance harmony in interpersonal relationships. Both Zhu and Wang could simply be advocating eliminating selfish desires for personal moral cultivation (for the self).
8. P. 9, Line 401: This statement here (“Neo-Confucianism took seriously the human right for survival”) should be placed with more emphasis as it directly challenges the statement made by MacIntyre. It could be used as the main thread for the whole paper to show either a transition in Confucian conception (from the suppression of human desires to the advocacy of the right to human desire) or as a refutation of MacIntyre’s claim.
Recommendation:
- Accept after Minor Revisions: The paper can in principle be accepted after revision based on the reviewer’s comments.
Author Response
Thank you so much for your carefulness to review the essay. Your comments and recommendations are considerable help to the author. Listed below please see the details of revisions:
Response General Comment#
Again sincere thanks to the reviewer for making an comprehensive and insightful general comment. It does be very necessary to explain the notion of “common principle” from the start, and correspondently a concise explanation of it has been added in line 100-104 of the track-change revision. As for how to respond to MacIntye’s claim, the author prefers the latter proposal the reviewer has mentioned, which has been added at the end of the part 6 (conclusion). With such a significant supplement, the theme of this essay appears more explicit and complete.
Response Specific Comments#
Point 1:
Changed the English rendition from Analects of Confucius in the original manuscript into an accurate one and added some justification to author’s criticism of traditional Confucianism as “subjectivism” in line 64-67 of the track-change revision.
Point2:
The key point of part 2 is to reveal what it means as an individual, and in what sense and to what extent individual limitations can be overcome. That is also why the author strengthens the point about Shao Yong’s recommended methodology “observing things on the basis of things.” As far as the extension of argument is concerned, the author does not think highlighting sage’s particularity in ability will weaken the key point of this part. In response to reviewer’s concern, the author has added a paragraph at the end of part 2 specially discussing this.
Point3:
Added a lot of explanations in page 5 of the track-change revision addressing the reviewer’s concerns on “how could any ethical principle be complete.”
Point4:
The reviewer is quite correct to recommend rephrasing in Lines 189-191 of the original manuscript, and the author approves this recommendation very much. Accordingly, a necessary adjustment has been made in 230-231 of the track-change revision.
Point5:
Changed “otherwise they find themselves in excessive” into “and not leave them alone casually” in line 267 of the track-change revision.
Point6:
Changed “inevitable” into “inescapable” in line 278 of the track-change revision.
Point7:
It is an excellent proposal that requires explaining how the mainstream Neo-Confucian “attached great importance to the harmony between individuals and community.” In view of Zhu Xi’s argument has been shown in line 228-229 of the original manuscript, the author has added Wang Yangming’s argument on this in 327-333 of the track-change revision.
Point8:
This is a significant recommendation also mentioned in the general comment by the reviewer, which has been implemented by adding related arguments at the end of the part 6 (conclusion).
Reviewer 2 Report
Dear Author:
Thanks for your contribution to the field! Your article clearly fulfills the goal stated at the beginning: to enrich the dialogue of global philosophies on individuality and community drawing upon Neo-Confucian sources. Therefore, its publication is warranted.
I have a few suggestions for revision. The first three are minor, but the last one is major.
1, Quotes from Analects should mark the chapter number (such as Analects 2:14) so that scholars have a universal referring system of Confucian classics.
2, The Quote of Wang Yangming in line 277-283 should be marked.
3, Alternative translations of 良知 are "conscientious knowing" or "good knowing." Alternative translations of 心 are "heartmind" or "mind-heart." Your translations of these terms make sense to me, so please just take these alternatives for consideration.
4, Wang Yangming’s advocacy on individual autonomy and independent free thought is worth emphasizing, particularly when Li Zhi’s thought was seen as a continuation of Wang’s thought. I would suggest to add one transitional paragraph at the end of part 4 or at the beginning of part 5 to present this aspect of Wang’s thought. Another benefit for doing this is to show that the so-called “mainstream” Neo-Confucian thought is not entirely contrary to what later reformative thinkers (such as Li Zhi and Wang Fuzhi in question) contributed to. Zhu Xi and Wang Yangming did not intend to eliminate all human desires when they urged to use heavenly principles to win over selfish desires. It increases the coherence of the article if this is pointed out.
Please do address the points of 1, 2 and 4 in your revision.
Best!
Author Response
Many thanks for your comments. In the light of your helpful suggestions, this essay has undergone some revisions which are very necessary. Details of those are as follows:
Response 1:
Marked the chapter number for the quotes from Analects of Confucius.
Response 2:
Marked the quote of Wang Yangming with a four-character indent in line 314-321 of the track-change revision.
Response 3:
Changed translations of 良知 into “conscientious knowing,” 心 into “mind-heart.”
Response 4:
I want to greatly appreciate the reviewer for making the suggestion to complement Wang Yangming’s advocacy on individual autonomy and independent free thought. The essay changed significantly in adding a paragraph focused on this issue at the end of part 4, which both strengthens the argument and increases the coherence of the article.